# Effects of Intergeneric Grafting of Schisandraceae on Root Morphology, Anatomy and Physiology of Rootstocks

Hong-Yi Liao [1,2], Sen Wang [1,2,*] and Chun-Yu Zhou [1,2]

1 Key Laboratory of Cultivation and Protection for Non-Wood Forest Trees of Ministry of Education, College of Forestry, Central South University of Forestry and Technology, Changsha 410004, China; lhy6511_21@163.com (H.-Y.L.); zhouchunyu2333@163.com (C.-Y.Z.)
2 The Belt and Road International Joint Research Center for Tropical Arid Non-Wood Forest in Hunan, Changsha 410004, China
* Correspondence: csuftwangsen@163.com

**Abstract:** *Kadsura coccinea* (Lem.) A. C. Smith and *Schisandra sphenantha* rehd. et wils. are different genus plants of Schisandraceae, distributed in the north and south of China, respectively. These species are non-wood forest trees with important medicinal value. Grafting is one of the technical means by which plants migrate to other regions to adapt to different habitats. However, the current research on the grafting of Schisandraceae only focuses on the effect of the rootstock on the scion, and lacks investigations on the effect of the scion on the rootstock, leading to the mechanism of the rootstock–scion interaction being still unclear. In this study, *Kadsura coccinea* (K) and *Schisandra sphenanthera* (S) were used as scion and rootstock for reciprocal grafting, and four grafted combinations, K/S (scion/rootstock), S/S, S/K, and K/K, were obtained, with S/S and K/K being the controls. Comparative analyses of the morphological, anatomical, and physiological characteristics of rootstocks were conducted at 10 d, 30 d, 50 d, and 70 d after grafting. The results showed that compared with the control, the total root length, root volume, root tips, and root biomass of K/S and S/K decreased. However, the root average diameter and root surface area of K/S increased, while those of S/K decreased. All the differences between the above indices and the control were about 10%, and almost all of them changed most significantly at 70 d after grafting. Anatomically, the root cross section, stele diameter, and xylem area of K/S increased, while those of S/K decreased. In addition, the cortex thicknesses of K/S and S/K increased. These anatomical changes were mainly reflected in the second-order and third-order roots. Meanwhile, the changes in root endogenous hormone (IAA, ZR, GA₃, ABA) contents in K/S and S/K in relation to days after grafting could explain the differences in root morphology. Moreover, both K/S and S/K had lower root activity and soluble sugar content than the control, and S/K had lower soluble protein and higher malondialdehyde content. This study indicated that the intergeneric grafting of Schisandraceae inhibited the growth of rootstocks, and the degree of inhibition was potentially related to the relative strength of the rootstock and scion, which provides a theoretical basis for further research on the rootstock–scion interaction mechanism.

**Keywords:** Schisandraceae; grafting; root morphology; anatomical structure; physiological characteristics

## 1. Introduction

    Schisandraceae is a dicotyledonous plant of Illicium Magnoliae, including the species *Schisandra* and *Kadsura* [1]. Medicinal plants make up the majority in this family, which have important systematic and medicinal research value [2]. *Kadsura coccinea* (Lem.) A. C. Smith, mainly distributed in southern China, is an evergreen woody vine of *Kadsura*, while *Schisandra sphenantha* rehd. et wils., mainly distributed in northern China, is a perennial deciduous woody vine of *Schisandra*; both of them are important medicinal resources, and their medicinal parts include their roots, vines, and fruits, which can treat many

diseases and have high medicinal and nutritional values [3–7]. At present, the research on Schisandraceae plants mostly focuses on medicinal components, germplasm resources, and morphological characteristics, with little research on reproduction and cultivation and even less research on grafting. The reports on grafting have only focused on the effects of distant grafting on scions [8].

Grafting, the process of attaching a shoot or a bud (scion) of one plant onto the stem or root of another plant (rootstock), is widely used not only to confer disease resistance and improve productivity under non-biological stress, but also as one of the means of realizing plant migration to other regions. After grafting, the scion and rootstock form a complete symbiotic entity and establish competitive and symbiotic relationships with each other [9]. The transport and distribution of photosynthates from scion to rootstock directly affect the growth of the rootstock, and the development of the rootstock also has feedback effects on the growth and development of the scion [10]. The growth and development of grafted plants depend on the coupling reactions between the scion and rootstock [11–13]. The study of the underlying mechanisms of scion–rootstock interactions is of great significance for revealing graft affinity, nutrient balance, and organ morphogenesis regulation [14]. Current studies on scion–rootstock interactions have mainly focused on graft affinity and the effects of the rootstock on the scion, while little research on the effects of the scion on the rootstock has been performed. Therefore, it is particularly important to explore the influence of the scion on the growth and physiological metabolism of the rootstock [15].

The growth and development of the rootstock depends on not only the moisture and mineral nutrients it absorbs but also the photosynthetic products and plant hormones supplied by the scion through the phloem [16]. The morphology, anatomy, and physiological characteristics of the rootstock are inevitably regulated by feedback from the scion [17], and different scions have different effects on the same rootstock [18], resulting in differences in the growth, physiological metabolism, and stress resistance of the graft [19,20]. Chang et al. [21] and Ding et al. [22] have shown that different scions can not only change the morphological index such as root length and root surface area but also affect vessels. Sélima Naija et al. [23] found that compared with a normal scion, a short-branch scion can make the rootstock sparser. Jiao et al. [24] also confirmed that when two different varieties of watermelon seedlings are inter-grafted as scion and rootstock, the effects of the scion on the growth and development of the rootstock vary depending on the grafted combination. Some studies have shown that compared with self-root seedlings, grafting can change physiological indices such as the root activity, POD activity, and soluble protein content of the rootstock [25,26], and its promoting or inhibitory effect depends on the rootstock–scion combination. The above studies mainly focus on the effects of grafting or interspecific grafting on rootstocks. So far, there are few reports about the effects of intergeneric grafting on rootstocks.

Therefore, this study measured the root morphology, anatomical structure, and physiological characteristics of rootstocks to evaluate the effects of scions with different genetic relationships on the rootstocks of Schisandraceae, with the aim of exploring the effects of intergeneric grafting on rootstocks of Schisandraceae and providing a theoretical basis for further research on the rootstock–scion interaction mechanism.

## 2. Materials and Methods

### 2.1. Plant Materials and Experimental Site

The seeds of *Kadsura coccinea* were collected from a plantation in Tongdao County, Huaihua City, Hunan Province, China, while the seeds of *Schisandra sphenanthera* were collected from Beiao Village, Luanchuan County, Luoyang City, Henan Province, China. Both of them were sourced from the same maternal tree. The experiment was conducted in the Key Laboratory of Cultivation and Protection for Non-Wood Forest Trees, Ministry of Education, Central South University of Forestry and Technology, Changsha, China (lat. 28°14′ N, long. 112°99′ E), from 2021 to 2022.

### 2.2. Experimental Design

The collected seeds of *Kadsura coccinea* and *Schisandra sphenanthera* were rinsed with tap water, soaked in 0.1% potassium permanganate solution for 15 min, and then buried in a sandbox (sand and water were mixed at 3:1 *v/v*) and placed in an artificial low-temperature chamber (4 °C), in which the humidity was maintained at 30% by covering the sandboxes with plastic wrap. Low temperature and appropriate humidity were conducive to breaking the dormancy of the seeds. The seeds of *Kadsura coccinea* were stored in sand two months later than *Schisandra sphenanthera* to ensure their germination at the same time. The germinated seeds were then transferred to seedling trays measuring 4 cm × 4 cm × 8 cm (length × width × height), filled with a mixture of peat (Klasmann, Geeste, Niedersachsen, Germany) and perlite at a ratio of 4:1 (*v/v*), and then moved to an artificial growth chamber (22–25 °C). When the first true leaf of the seedling was unfolded (the stem diameter, leaf area and leaf biomass of *Schisandra sphenanthera* were about 1 cm, 0.5 cm$^2$, and 5 mg, respectively, while those of *Kadsura coccinea* were about 2 cm, 2.5 cm$^2$, and 20 mg, respectively) (Figure 1A), grafting was carried out using the approach grafting method [27], and the grafted seedlings were transplanted into a cylindrical seedling-raising pot (peat and perlite mixed at 4:1 *v/v*) with a diameter of 10 cm at the bottom and a height of 15 cm. There were four grafted combinations (Figure 1B, Table 1). Treatments 1 and 3 were heterologous graftings and designated as the experimental group, with a graft survival rate of about 60%; treatments 2 and 4 were homologous graftings and designated as the controls, with a graft survival rate of about 80%. At 10 d, 30 d, 50 d, and 70 d after grafting, grafted seedlings with similar growth performance in each treatment were randomly selected as samples. All experiments were sampled with 10 seedlings per treatment and repeated three times.

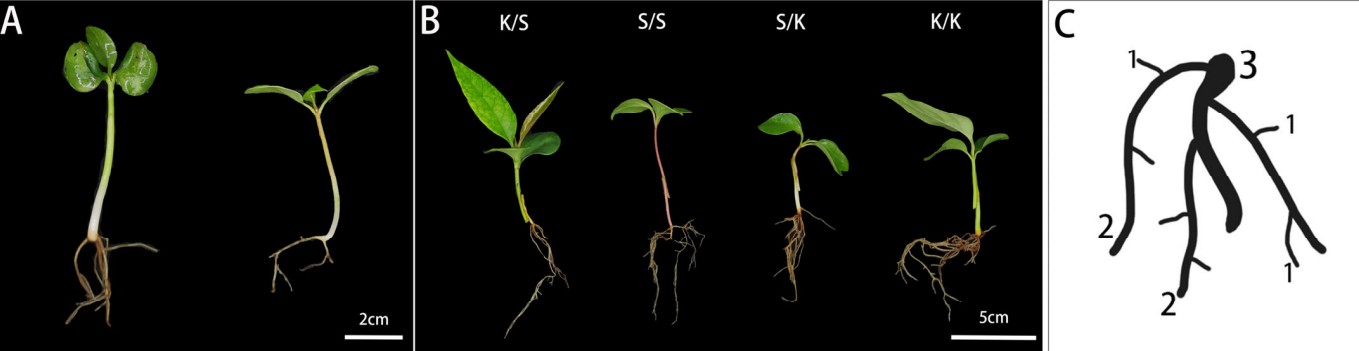

**Figure 1.** (**A**) The first true leaf of *Kadsura coccinea* (left) and *Schisandra sphenanthera* (right) seedlings unfolding. (**B**) The grafted combinations of K/S, S/S, S/K, K/K. (**C**) Schematic diagram of the root order grading method. The numbers 1/2/3 refer to the first-, second-, and third-order roots respectively.

**Table 1.** Design of scion and rootstock combinations.

| Treatment | Scion | Rootstock | Grafted Combination |
|:---:|:---:|:---:|:---:|
| 1 | K | S | K/S |
| 2 | S | S | S/S |
| 3 | S | K | S/K |
| 4 | K | K | K/K |

Note: K is *Kadsura coccinea* seedling, S is *Schisandra sphenanthera* seedling. Treatments 2 and 4 are the controls.

### 2.3. Root Morphology and Biomass

After sampling, the entire root of each seedling was cut at the root–stem junction using a blade, then washed with tap water to remove soil. After rinsing, the roots were blotted dry with filter paper. All root samples were scanned using a scanner (Epson Scan, Japan), and the root analysis software (WinRHIZO PRO 2013, Regent Instruments, Quebec,

QC, Canada) was used to analyze the total root length (TRL, cm), root surface area (RSA, $cm^2$), root average diameter (RAD, mm), root volume (RV, $cm^3$), and root tips (RT) for each sample. After scanning, the root samples were dried in a 60 °C oven for 72 h to a constant weight, and the dry weight was measured using an electronic balance to obtain root biomass (RB, g).

### 2.4. Root Anatomical Structure

According to the root order grading method [28] (Figure 1C), the parts above the root tip of the first-, second-, and third-order roots of the rootstock were excised with a blade. Each root segment was cut 5 mm long, and 6 root segments were randomly selected for each grafted combination. The root segments were immediately fixed in FAA (5 mL 37% formaldehyde, 5 mL glacial acetic acid, 90 mL 50% ethanol) for at least 24 h. The sections were made using the paraffin section method, which involved a series of steps including dehydration in alcohol with progressively increasing concentration, clarification with xylene, embedding in paraffin, section making, section sticking, deparaffinization, staining, and section sealing [29], and each section is 10 μm. The sections were then observed and photographed using an optical microscope (BX-51, Olympus, Tokyo, Japan). The root cross-section (mm), cortex thickness (mm), stele diameter (mm), and xylem area ($mm^2$) were measured using ImageJ v1.8.0.112 software.

### 2.5. Root Endogenous Hormone Content

The endogenous hormone content in roots were determined by the enzyme-linked immunosorbent assay (ELISA) [30]. In liquid nitrogen, 0.5 g of fresh root samples (a mixture of root segments from different grafted seedlings) were ground into fine powder, followed by 2 mL of sample extraction solution, which was ground into homogenate in an ice bath and transferred to a 10 mL test tube. The mortar was washed with 2 mL of extraction solution, which was then added to the same test tube. The solution was thoroughly shaken and stored at 4 °C for 4 h. Afterward, the sample was centrifuged at 3500 rpm for 8 min, and the supernatant was collected. The residue was extracted again with 1 mL of extraction solution, shaken, and centrifuged at 4 °C for 1 h to obtain the supernatant. The two supernatants were combined, recorded in volume, and purified on a C18 Sep-Pak column (Waters, Milford, MA, USA) to obtain an extract, which was then dried under nitrogen. The residue was dissolved in 0.01 M pH 7.5 phosphate-buffered saline (PBS) for the measurement of IAA, ZR, $GA_3$, and ABA.

### 2.6. Root Activity

Root activity was determined using the triphenyltetrazolium chloride (TTC) reduction method [31]. About 0.2 g of fresh root samples were placed in a 15 mL calibrated test tube, and the root samples were completely immersed in an equal volume solution of 1% TTC and 0.1 mol/L PBS. Test tubes were then kept at 37 °C for 1 h. The reaction was terminated by adding 2 mL of 1 mol/L sulfuric acid. Finally, the root samples were transferred to a mortar and ground with 10 mL of ethyl acetate. After centrifugation, the supernatant was measured at a wavelength of 485 nm using the colorimetric method.

### 2.7. Root Soluble Sugar and Malondialdehyde Content

The contents of soluble sugars and malondialdehyde (MDA) were determined using the thiobarbituric acid (TBA) heating method [31]. About 0.2 g of fresh root samples was ground with 10% trichloroacetic acid (TCA) to form a homogenous mixture and centrifuged at 4000 rpm for 10 min. Then, 2 mL of the supernatant was mixed with 2 mL of 0.6% TBA and boiled in a hot water bath until the color changed. After rapid cooling, the mixture was centrifuged at 4000 rpm for 10 min. The content of soluble sugar and MDA were calculated by measuring the supernatant colorimetrically at 450 nm, 532 nm, and 600 nm.

### 2.8. Root Soluble Protein Content

The soluble protein content was determined using the Coomassie brilliant blue method [31]. About 0.2 g of fresh root samples was mixed with pH 7.8 PBS, ground in an ice bath, and transferred to a centrifuge test tube. The mixture was then centrifuged at 4500 rpm for 10 min. Next, 0.1 mL of supernatant was mixed with 0.9 mL of water and 5 mL of Coomassie brilliant blue G-250 reagent and thoroughly shaken. After standing for 2 min, absorbance was recorded at 595 nm.

### 2.9. Statistical Analysis

The data were analyzed using one-way analysis of variance (ANOVA) and mean comparison, and the differences were statistically compared by employing the Duncan test with a significance level of $p < 0.05$. Data are expressed as mean $\pm$ standard error (SE). All statistical analyses were carried out using the SPSS software version 21.0 (SPSS Inc., Chicago, IL, USA). The figures were prepared using Origin 2021 (Origin Laboratory, Northampton, MA, USA).

## 3. Results

### 3.1. Root Morphology and Biomass of Different Grafted Combinations

Through observing the root morphology of the grafted S/S, K/S, K/K, and S/K seedlings at each stage after grafting, we found that the intergeneric grafting changed the root growth of the rootstock ($p < 0.05$) (Figure 2,3). Compared with the control (S/S, K/K), K/S showed a decrease in TRL, RV, RT, and RB and an increase in RAD at 10 d, 30 d, 50 d, and 70 d after grafting, while all indices of S/K decreased. At 30 d after grafting, the TRL of K/S was significantly lower than S/S by about 10% (Figure 3A). At 70 d after grafting, the RAD of K/S was significantly larger than S/S by about 5% (Figure 3B), but the RV, RT, and RB of K/S were significantly reduced compared to S/S by 6.8%, 12.4%, and 13.6%, respectively (Figure 3D–F). Meanwhile, the TRL, RSA, RAD, RT, and RB of S/K were all significantly lower than K/K at 70 d, with reductions of 3.3%, 2.6%, 7.7%, 11.8%, and 11.3%, respectively (Figure 3G–I,K,L), but there was no significant difference in RV between S/K and K/K (Figure 3K).

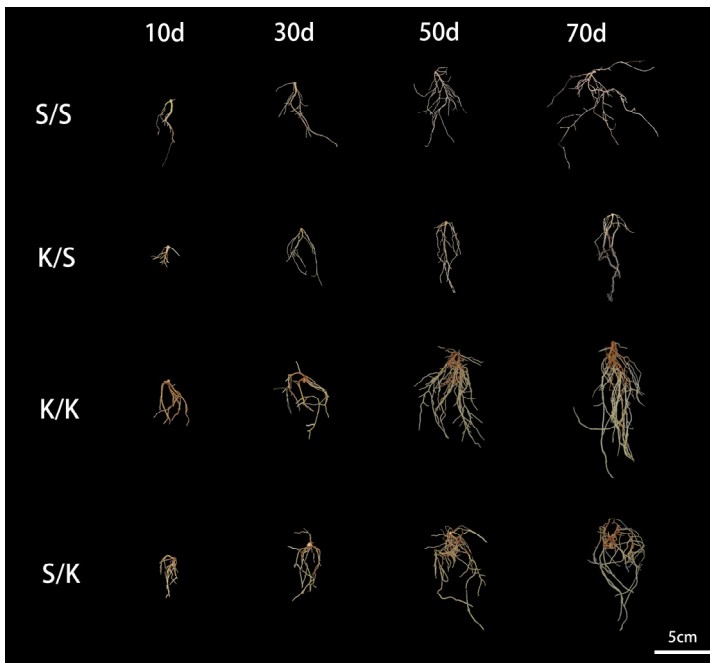

**Figure 2.** Root morphology observation of grafted S/S, K/S, K/K, and S/K seedlings.

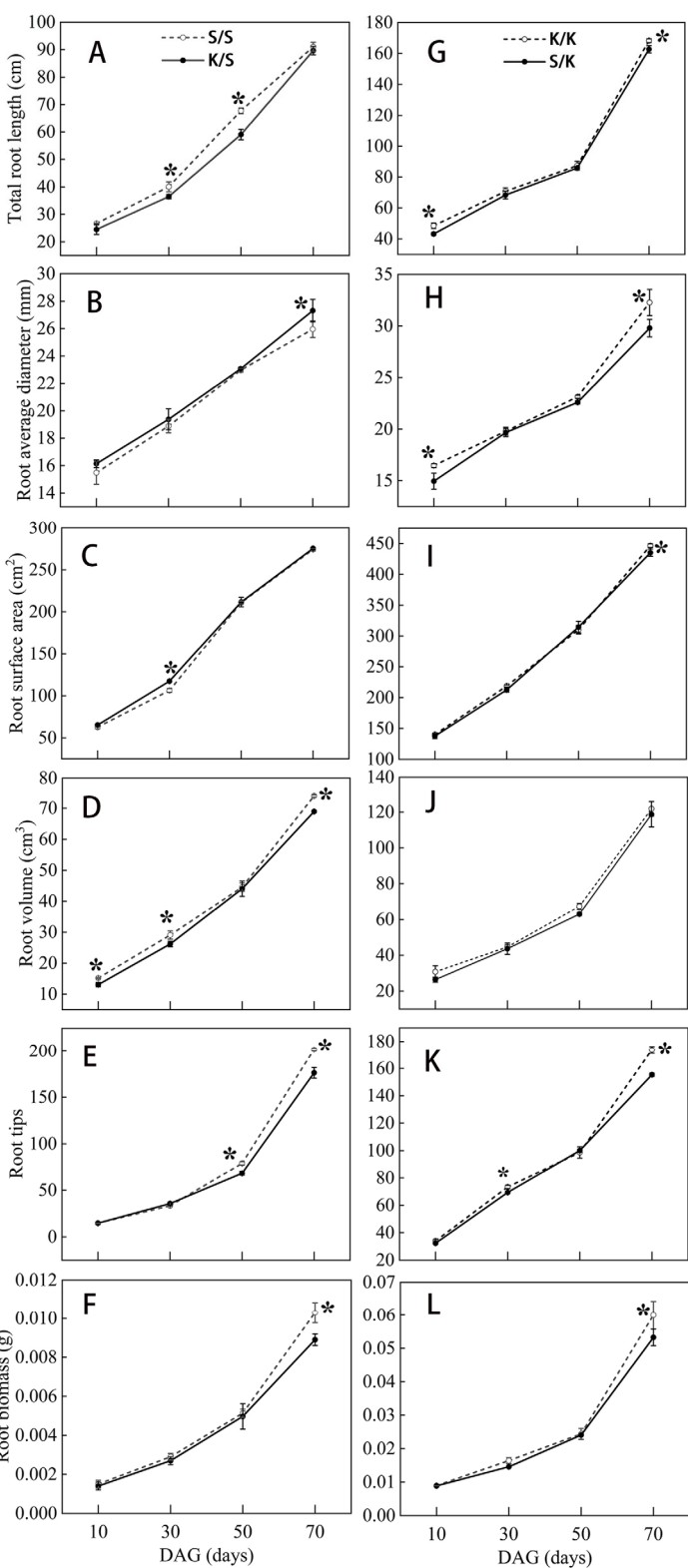

**Figure 3.** Root morphological indices of grafted S/S, K/S, K/K, and S/K seedlings at different grafting stages: (**A**,**G**) total root length (TRL), (**B**,**H**) root average diameter (RAD), (**C**,**I**) root surface area (RSA), (**D**,**J**) root volume (RV), (**E**,**K**) root tips (RT), (**F**,**L**) root biomass (RB). DAG: days after grafting. Error bars represent standard deviations. Data are means ± SE ($n = 10$). "*" indicates significant difference ($p < 0.05$) between grafted combinations under the same DAG.

### 3.2. Root Anatomy of Different Grafted Combinations

The root anatomical structures of each grafted combination at 10 d, 30 d, 50 d, and 70 d after grafting were observed, and the results showed that intergeneric grafting had effects on the anatomical structures of the first-, second-, and third-order roots of rootstocks ($p < 0.05$) (Tables 2 and 3; Figures 4 and 5). Compared with the control, the root cross-section, stele diameter, and xylem area of the first-order roots of K/S were increased, while cortex thickness was decreased. Only the change in stele diameter was significant, which decreased by about 0.04 mm at each stage. The second- and third-order roots of K/S showed consistent changes, with both having significantly larger root cross-sections, cortex thicknesses, stele diameters, and xylem areas than S/S at each stage. The average growth rates of the root cross-section and cortex thickness of the second-order root (15.1%, 18.5%) were smaller than those of the third-order root (21.5%, 28.5%), but the average growth rates of the stele diameter and xylem area of the second-order root (32.1%, 80.6%) were larger than those of the third-order root (15.1%, 30.9%) (Table 2). Meanwhile, compared with the control, the root cross-section, cortex thickness, and xylem area of the first-order root of S/K all increased, while stele diameter decreased, and they were significantly different from K/K at 10 d after grafting, with increases of 30.9%, 91.3%, and 64.3% and a decrease of 68%, respectively. The changes in the second- and third-order roots of S/K were consistent, with both having smaller root cross-sections and stele diameters and larger cortex thicknesses than K/K, and the root cross-sections of both were significantly different from K/K at 70 d after grafting, with decreases of 8.3% and 6.2%, respectively. In addition, for stele diameter, xylem area, and cortex thickness, the second-order root of S/K showed the largest difference from K/K at 30 d after grafting, with significant decreases of 24% and 47.5% and an increase of 18.7%, respectively; the third-order root of S/K showed the largest difference from K/K at 10 d after grafting, with significant decreases of 24.5% and 51.9% and an increase of 34.3%, respectively (Table 3).

**Table 2.** Anatomical structural parameters of the first-, second-, and third-order roots of grafted K/S and S/S seedlings at different grafting stages.

| Root Order | DAG (d) | Grafted Combination | Root Cross-Section (mm) | Cortex Thickness (mm) | Stele Diameter (mm) | Xylem Area (mm²) |
|---|---|---|---|---|---|---|
| First | 10 | S/S | 0.1839 ± 0.0178 a | 0.0780 ± 0.0125 a | 0.0278 ± 0.0071 a | 0.0005 ± 0.0001 a |
| | | K/S | 0.2350 ± 0.0164 b | 0.0834 ± 0.0018 a | 0.0681 ± 0.0128 b | 0.0007 ± 0.0001 a |
| | 30 | S/S | 0.2744 ± 0.0184 a | 0.1108 ± 0.0131 b | 0.0528 ± 0.0086 a | 0.0013 ± 0.0002 a |
| | | K/S | 0.2840 ± 0.0155 a | 0.0933 ± 0.0020 a | 0.0974 ± 0.0116 b | 0.0015 ± 0.0002 a |
| | 50 | S/S | 0.3956 ± 0.0228 a | 0.1615 ± 0.0030 a | 0.0725 ± 0.0168 a | 0.0026 ± 0.0001 a |
| | | K/S | 0.4054 ± 0.0153 a | 0.1446 ± 0.0116 a | 0.1161 ± 0.0078 b | 0.0032 ± 0.0001 b |
| | 70 | S/S | 0.4631 ± 0.0164 a | 0.1910 ± 0.0092 a | 0.0811 ± 0.0027 a | 0.0030 ± 0.0001 a |
| | | K/S | 0.4878 ± 0.0116 a | 0.1811 ± 0.0054 a | 0.1256 ± 0.0223 b | 0.0042 ± 0.0002 b |
| Second | 10 | S/S | 0.2577 ± 0.0096 a | 0.0808 ± 0.0055 a | 0.0661 ± 0.0094 a | 0.0016 ± 0.0001 a |
| | | K/S | 0.3149 ± 0.0116 b | 0.1094 ± 0.0064 b | 0.0806 ± 0.0029 b | 0.0012 ± 0.0003 a |
| | 30 | S/S | 0.2882 ± 0.0054 a | 0.1099 ± 0.0069 a | 0.0984 ± 0.003 a | 0.001 ± 0.0001 a |
| | | K/S | 0.3387 ± 0.0112 b | 0.1199 ± 0.0042 b | 0.1189 ± 0.0033 b | 0.0032 ± 0.0001 b |
| | 50 | S/S | 0.364 ± 0.0051 a | 0.1102 ± 0.0052 a | 0.1436 ± 0.0048 a | 0.0037 ± 0.0001 a |
| | | K/S | 0.3963 ± 0.0058 b | 0.1263 ± 0.0024 b | 0.187 ± 0.0096 b | 0.0075 ± 0.0004 b |
| | 70 | S/S | 0.4449 ± 0.0112 a | 0.112 ± 0.0088 a | 0.2209 ± 0.0072 a | 0.0102 ± 0.0005 a |
| | | K/S | 0.499 ± 0.0045 b | 0.1287 ± 0.0077 b | 0.2415 ± 0.0179 b | 0.0127 ± 0.0003 b |
| Third | 10 | S/S | 0.2936 ± 0.0014 a | 0.1068 ± 0.0007 a | 0.0801 ± 0.0003 a | 0.0013 ± 0.0001 a |
| | | K/S | 0.3836 ± 0.0086 b | 0.1402 ± 0.0035 b | 0.1031 ± 0.002 b | 0.0016 ± 0.0001 a |
| | 30 | S/S | 0.4994 ± 0.0117 a | 0.1596 ± 0.0048 a | 0.1802 ± 0.0023 a | 0.0056 ± 0.0003 a |
| | | K/S | 0.6721 ± 0.0056 b | 0.2266 ± 0.0067 b | 0.2189 ± 0.0087 b | 0.0102 ± 0.0002 b |
| | 50 | S/S | 0.6696 ± 0.0117 a | 0.1524 ± 0.0107 a | 0.3648 ± 0.0152 a | 0.0475 ± 0.0048 a |
| | | K/S | 0.7219 ± 0.002 b | 0.1757 ± 0.0007 b | 0.3706 ± 0.0032 a | 0.0535 ± 0.0031 b |
| | 70 | S/S | 0.7349 ± 0.0097 a | 0.0934 ± 0.0034 a | 0.548 ± 0.0042 a | 0.1066 ± 0.0048 a |
| | | K/S | 0.8298 ± 0.0091 b | 0.1172 ± 0.0092 b | 0.5954 ± 0.0253 b | 0.1127 ± 0.0055 b |

Note: DAG: days after grafting. Means provided with standard errors. Different lowercase letters following the values indicate significant difference ($p < 0.05$) between grafted combinations under the same DAG.

**Table 3.** Anatomical structural parameters of the first-, second-, and third-order roots of grafted S/K and K/K seedlings at different grafting stages.

| Root Order | DAG (d) | Grafted Combination | Root Cross-Section (mm) | Cortex Thickness (mm) | Stele Diameter (mm) | Xylem Area (mm$^2$) |
|---|---|---|---|---|---|---|
| First | 10 | K/K | 0.2085 ± 0.01894 a | 0.0647 ± 0.0028 a | 0.0790 ± 0.0134 a | 0.0014 ± 0.0001 a |
| | | S/K | 0.2729 ± 0.00539 b | 0.1238 ± 0.0062 b | 0.0253 ± 0.0073 b | 0.0023 ± 0.0002 b |
| | 30 | K/K | 0.3435 ± 0.02008 a | 0.1320 ± 0.0049 a | 0.0795 ± 0.0038 a | 0.0018 ± 0.0001 a |
| | | S/K | 0.3581 ± 0.0211 a | 0.1427 ± 0.0121 a | 0.0727 ± 0.0041 a | 0.0028 ± 0.0002 b |
| | 50 | K/K | 0.5080 ± 0.0191 a | 0.1767 ± 0.0230 a | 0.1546 ± 0.0275 a | 0.0055 ± 0.0005 a |
| | | S/K | 0.5501 ± 0.0207 a | 0.2018 ± 0.0172 a | 0.1465 ± 0.0142 a | 0.0041 ± 0.0003 a |
| | 70 | K/K | 0.6707 ± 0.0267 a | 0.2204 ± 0.0333 a | 0.2299 ± 0.0402 a | 0.0063 ± 0.0003 a |
| | | S/K | 0.7717 ± 0.0130 b | 0.2785 ± 0.0193 a | 0.2148 ± 0.0257 a | 0.0159 ± 0.0029 b |
| Second | 10 | K/K | 0.4591 ± 0.0174 a | 0.126 ± 0.0122 a | 0.1672 ± 0.035 a | 0.0044 ± 0.0019 a |
| | | S/K | 0.4278 ± 0.014 b | 0.1393 ± 0.0098 a | 0.1492 ± 0.0073 a | 0.0026 ± 0.0001 a |
| | 30 | K/K | 0.4902 ± 0.0043 a | 0.1378 ± 0.0032 a | 0.2146 ± 0.003 a | 0.0086 ± 0.0003 a |
| | | S/K | 0.4601 ± 0.0142 b | 0.1635 ± 0.0123 b | 0.1631 ± 0.031 b | 0.0045 ± 0.0044 b |
| | 50 | K/K | 0.5521 ± 0.0191 a | 0.1589 ± 0.0093 a | 0.2342 ± 0.0045 a | 0.0105 ± 0.0003 a |
| | | S/K | 0.5182 ± 0.0091 b | 0.1689 ± 0.0056 a | 0.1805 ± 0.0026 b | 0.0056 ± 0.0002 b |
| | 70 | K/K | 0.6002 ± 0.0004 a | 0.1588 ± 0.0052 a | 0.2785 ± 0.002 a | 0.0136 ± 0.0003 a |
| | | S/K | 0.5502 ± 0.0095 b | 0.1808 ± 0.001 b | 0.2726 ± 0.0102 a | 0.0107 ± 0.0007 b |
| Third | 10 | K/K | 0.7603 ± 0.0133 a | 0.1556 ± 0.004 a | 0.4491 ± 0.0102 a | 0.0762 ± 0.0006 a |
| | | S/K | 0.757 ± 0.0315 a | 0.209 ± 0.0186 b | 0.339 ± 0.0065 b | 0.0367 ± 0.0014 b |
| | 30 | K/K | 0.8577 ± 0.0071 a | 0.082 ± 0.0036 a | 0.6936 ± 0.0113 a | 0.1439 ± 0.0057 a |
| | | S/K | 0.8885 ± 0.0148 a | 0.1035 ± 0.0107 a | 0.6816 ± 0.0089 a | 0.1222 ± 0.0063 b |
| | 50 | K/K | 0.9818 ± 0.0075 a | 0.0632 ± 0.0044 a | 0.8554 ± 0.0078 a | 0.2165 ± 0.0019 a |
| | | S/K | 0.9662 ± 0.0149 a | 0.067 ± 0.0223 a | 0.8321 ± 0.0319 a | 0.2067 ± 0.0066 b |
| | 70 | K/K | 1.2555 ± 0.0137 a | 0.0791 ± 0.0072 a | 1.0973 ± 0.0091 a | 0.4237 ± 0.0066 a |
| | | S/K | 1.1775 ± 0.0139 b | 0.0625 ± 0.0104 a | 1.0525 ± 0.0154 b | 0.3793 ± 0.0067 b |

Note: DAG: days after grafting. Means provided with standard errors. Different lowercase letters following the values indicate significant difference ($p < 0.05$) between grafted combinations under the same DAG.

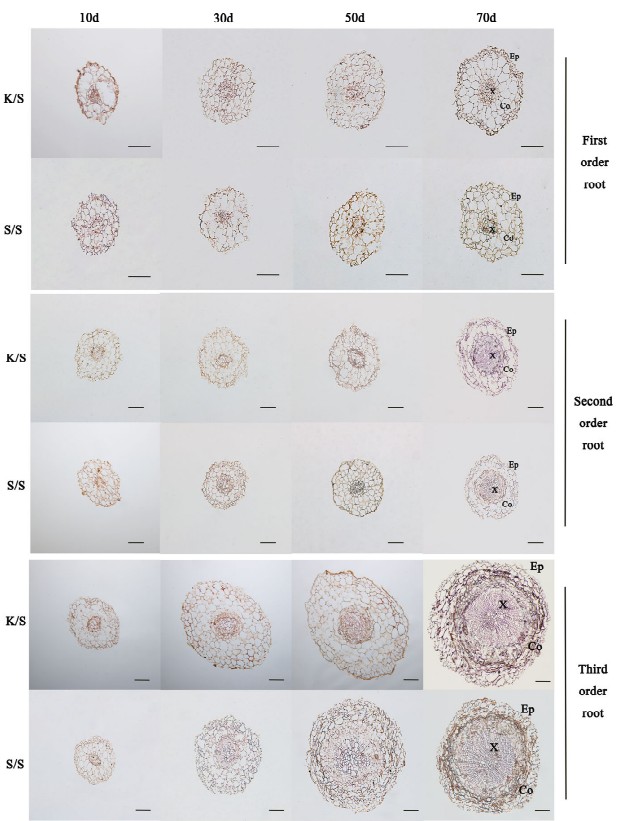

**Figure 4.** Anatomical structure observations of the first-, second-, and third-order roots of grafted K/S and S/S seedlings at different grafting stages. Ep: epidermis; Co: cortex; X: xylem. Scale bar = 100 μm.

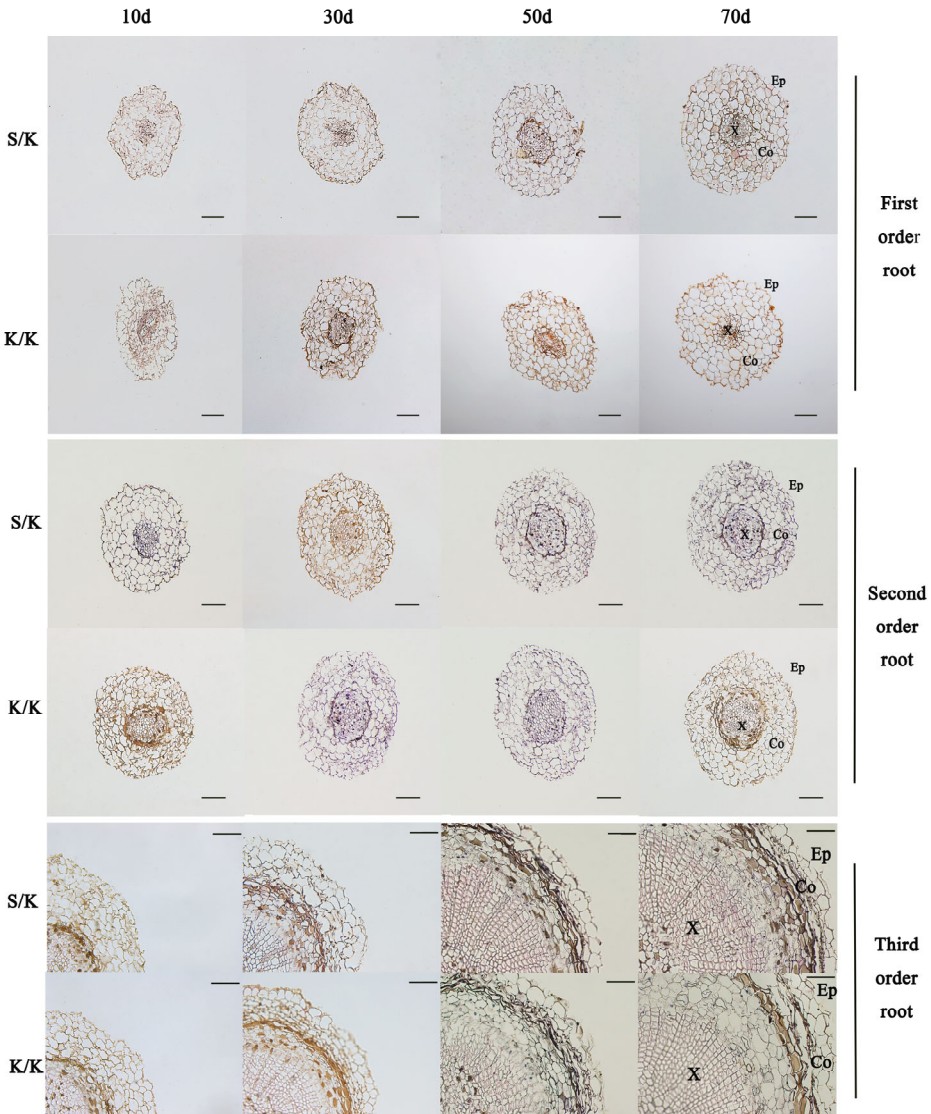

**Figure 5.** Anatomical structure observations of the first-, second-, and third-order roots of grafted S/K and K/K seedlings at different grafting stages. Ep: epidermis; Co: cortex; X: xylem. Scale bar = 100 μm.

### 3.3. Root Endogenous Hormone Contents of Different Grafted Combinations

Intergeneric grafting significantly affected the endogenous hormone content of root-stocks at different stages ($p < 0.05$) (Figure 6). The IAA content of K/S was consistently lower than S/S from 10 d to 50 d after grafting, but significantly higher than S/S at 70 d, increasing by 49.4% (Figure 6A), and its ZR content was significantly lower than that of S/S at every stage after grafting (Figure 6B). Conversely, the GA$_3$ and ABA contents of K/S were consistently higher than S/S, with the former showing the greatest significant difference at 70 d, increasing by 104.1% (Figure 6C), while the latter showed no significant difference from S/S at 70 d (Figure 6D). The IAA content of S/K was consistently significantly lower than K/K, with the greatest difference at 70 d, decreasing by 56.8% (Figure 6E), when the ZR content also showed the most significant difference between S/K and K/K, with a decrease of 30.3% (Figure 6F). However, the GA$_3$ content of S/K was significantly increased from 10 d to 50 d compared to K/K but significantly decreased at 70 d by 22.2% (Figure 6G). The ABA content of S/K was consistently higher than K/K (Figure 6H).

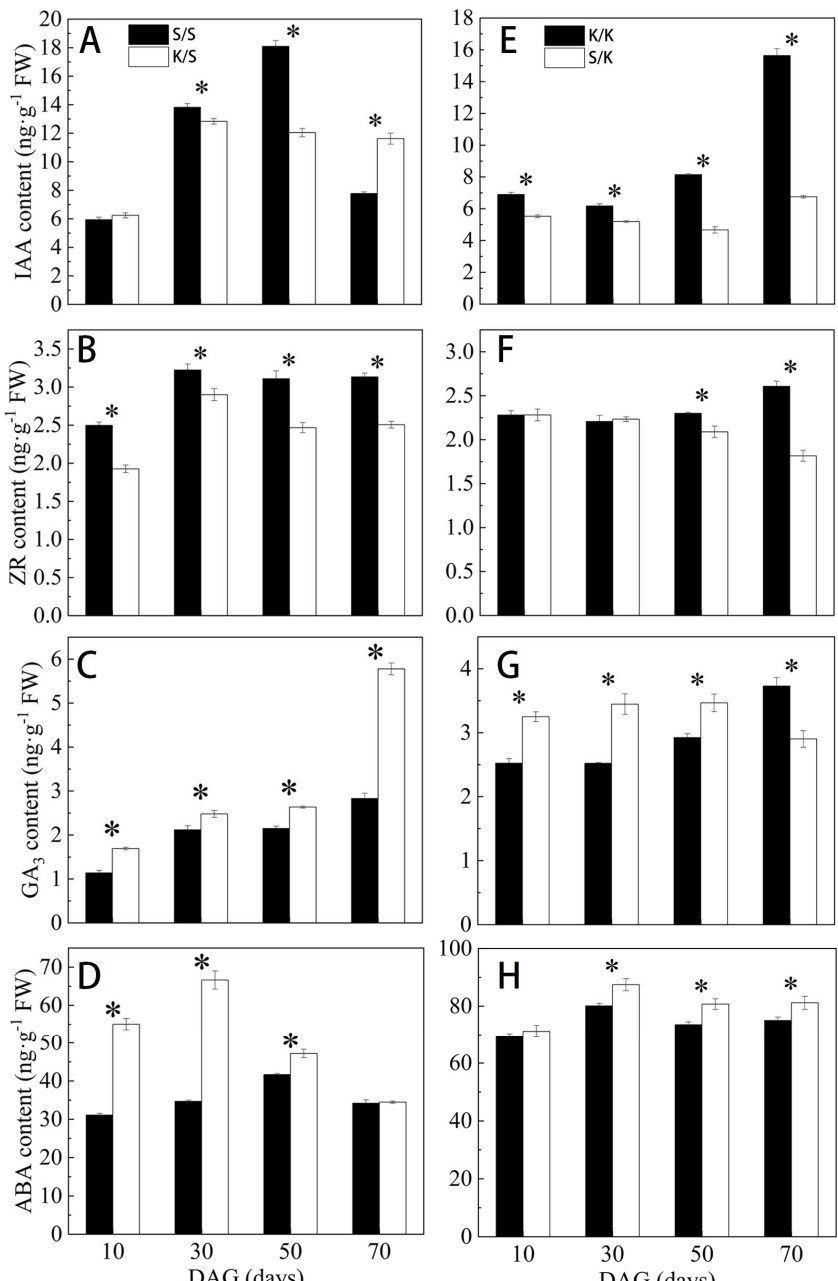

**Figure 6.** Root endogenous hormone content of grafted S/S, K/S, K/K, and S/K seedlings at different grafting stages: (**A**,**E**) IAA content, (**B**,**F**) ZR content, (**C**,**G**) GA$_3$ content, (**D**,**H**) ABA content. DAG: days after grafting. Error bars represent standard deviations. Data are means $\pm$ SE (*n* = 10). "*" indicates significant difference (*p* < 0.05) between grafted combinations under the same DAG.

### 3.4. Root Activity and MDA Content of Different Grafted Combinations

As shown in Figure 7, at each stage after grafting, the root activities of K/S and S/K were consistently lower than the control, with a significant reduction of 51.1% and 29.5% at 30 d and 50 d for K/S (Figure 6A) and 52.9%, 51.2%, and 46.4% at 30 d, 50 d, and 70 d for S/K (Figure 6C), respectively. However, the MDA content of K/S did not show a significant difference compared to S/S (Figure 6B), while the MDA content of S/K was significantly higher than that of K/K at 10 d, 30 d, and 70 d, with an increase of 174.08%, 17.4%, and 83.6%, respectively (Figure 6D).

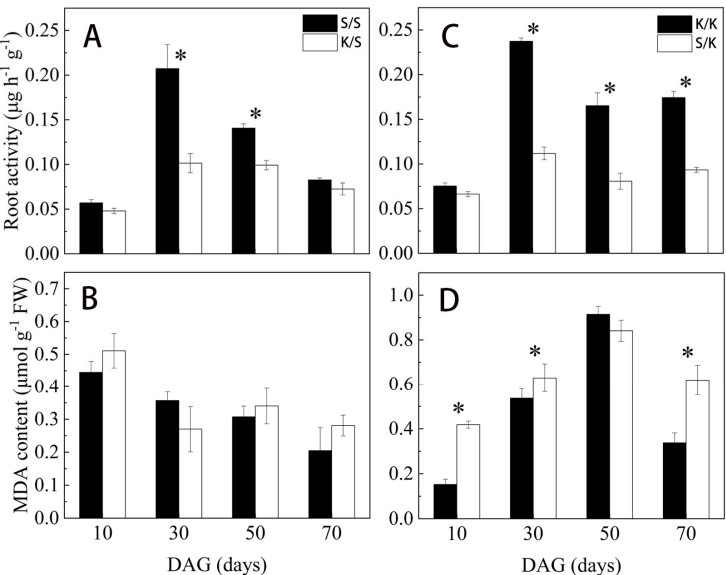

**Figure 7.** Root activity and MDA content of grafted S/S, K/S, K/K, and S/K seedlings at different grafting stages: (**A**,**C**) root activity, (**B**,**D**) MDA content. DAG: days after grafting. Error bars represent standard deviations. Data are means ± SE (*n* = 10). "*" indicates significant difference (*p* < 0.05) between grafted combinations under the same DAG.

*3.5. Root Soluble Sugar and Soluble Protein Content of Different Grafted Combinations*

As shown in Figure 8, the soluble sugar content of both K/S and S/K was consistently lower than the control at each stage after grafting, and the former was significantly lower than S/S at 30 d and 50 d by 26% and 18.7%, respectively; the latter showed significant differences with K/K at each stage, with the greatest difference at 10 d, decreasing by 41.6%. The soluble protein content of K/S showed no significant difference from S/S, while that of S/K was significantly lower than K/K at 10 d and 30 d, decreasing by 11% and 3.9%, respectively.

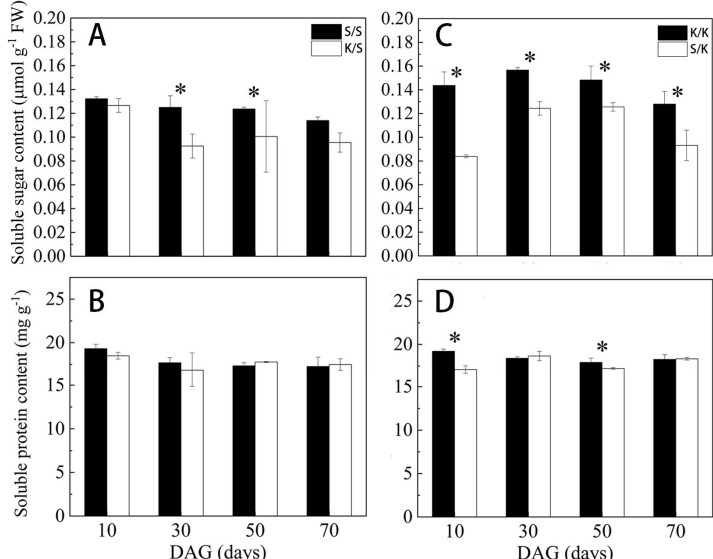

**Figure 8.** Soluble sugar and protein content of grafted S/S, K/S, K/K, and S/K seedlings at different grafting stages: (**A**,**C**) soluble sugar content, (**B**,**D**) soluble protein content. DAG: days after grafting. Error bars represent standard deviations. Data are means ± SE (*n* = 10). "*" indicates significant difference (*p* < 0.05) between grafted combinations under the same DAG.

## 4. Discussion

### 4.1. Effects of Intergeneric Grafting on Morphological Characteristics of Rootstocks

In this study, the comparative analysis of root morphological characteristics of four grafted combinations of Schisandraceae revealed significant differences among the combinations. At each stage after grafting, the total root length, root surface area, root average diameter, root volume, root tips, and root biomass of S/K were significantly reduced. Similar results were obtained by Zhao et al. in their study of watermelon-grafted seedlings, which showed that compared to pumpkin rootstock, watermelon as a heterologous scion significantly reduced the root dry weight and root volume of pumpkin rootstock, indicating that heterologous grafting had an inhibitory effect on the root growth of rootstocks. This was also consistent with previous studies on walnut, grape, citrus, peach, and other fruit trees [32]. In contrast, Suchoff et al. found that compared to the self-grafted control, the total root length and root tips of grafted tomato seedlings significantly increased in greenhouse pot experiments [33]. Kakita et al. also reported a similar increase in the root surface area of grafted tomato [34]. In addition, Huang et al. found that the root volume and root surface area of grafted watermelons were higher than those of non-grafted watermelons [35]. These studies have highlighted the enhancement effect of grafting, contrary to the results in this study. The inhibition or enhancement effect brought by this grafting on the rootstock might be related to the grafting affinity. The farther the genetic relationship between the rootstock and scion was, the lower the grafting affinity and the greater the inhibitory effect on root growth of rootstocks. *Kadsura coccinea* and *Schisandra sphenanthera*, which belong to different genera in Schisandraceae, have a relatively distant relationship, and their grafting affinity is definitely lower than that of self-grafting, resulting in poor growth of rootstocks. Similar to S/K, K/S had significantly lower total root length, root volume, root tips, and root biomass than the control S/S. However, the root average diameter of K/S was significantly increased, which was in complete contrast to the behavior of S/K. Studies have shown that the increase in root diameter can be used as a defense strategy by plants against the environment it is exposed to [36], or it can be triggered by the stimulation of cell division [37], but this change in K/S needs to be further verified by related experiments in a later stage.

### 4.2. Effect of Intergeneric Grafting on Anatomical Structure of Rootstock

Root anatomical characteristics not only directly represent the development of plant tissues but are closely related to their functions [38]. However, there are few studies that examine the anatomical and functional changes in roots based on their branching order. *Kadsura coccinea* and *Schisandra sphenanthera* are woody lianas, and their roots are fibrous roots, which usually have strong resource absorption capacity [39,40]. Wang et al. found through observations of cortical tissue that the first- and second-order roots of three lianas, including Schisandra chinensis, were the main sources of resource absorption [41]. Guo et al. studied the first- to fifth-order roots of 23 temperate woody plants in China and found that first-order roots had primary growth and were typical absorptive roots, while the second- and third-order roots were likely to be mainly absorbing or related to transportation and support, depending on the species [42]. In this study, the root development of the grafted seedlings reached grade 3–4 at 70 days after grafting. Therefore, the anatomical structures of the first-, second-, and third-order roots of four grafted combinations were studied, all of which were composed of epidermis, cortex, and stele.

This study found that, compared to the control, the root cross-sections of the first-, second-, and third-order roots of K/S increased, while the root cross-section of the first-order root of S/K increased and the root cross-section of the second- and third-order roots of S/K decreased, corresponding to the increase in the average root diameter of K/S and the decrease in the average root diameter of S/K. This also indicated that the thickening of the root in response to *Kadsura coccinea* scion grafting occurred in the first-, second-, and third-order roots of the *Schisandra sphenanthera* rootstock, while the refinement of *Kadsura coccinea* rootstock caused by *Schisandra sphenanthera* scion only occurred in the

second- and third-order roots. The diameter of the root cross-section is determined by cortex thickness and stele diameter [43]. Cortex thickness can affect the growth and physiological activities of plants, such as changing water resistance [44,45], respiratory intensity, and resource absorption [46,47]. The stele in the root includes conduits forming main conductive tissues. The size of the stele in the root cross-section is mainly used as an index of the transport potential of water and nutrients from root to shoot [42,48]. The thinner the cortex is, the stronger the lateral transport function of the root is, and the larger the diameter of the stele is, the stronger the longitudinal transport function of the root is [7]. In this study, the cortex thickness and stele diameter of K/S varied significantly in the second- and third-order roots, and both increased; meanwhile, the cortex thickness and stele diameter of S/K increased and decreased significantly in each root order, respectively. This indicated that intergeneric grafting weakened the lateral transportation capacity of the rootstock, but the changes in its longitudinal transportation capacity were related to different combinations of rootstock and scion. This might be because the growth structure of *Kadsura coccinea* is superior to that of *Schisandra sphenanthera*. A scion with higher capacity can better and more synthesize photosynthetic products and plant hormones to be transported to the rootstock [8] because of its larger leaves and thicker stems. Xylem is the main component of the stele, and its degree of development can increase the proportion of transporting tissues to improve water transport efficiency [49]. In this study, the xylem areas of each order roots of K/S increased, while those of S/K decreased in the second- and third-order roots, similar to the changes in stele diameter, which again verified the changes in its longitudinal transport capacity. These above changes also indicated that the effect of intergeneric grafting on the anatomical structure of the rootstock's second- and third-order roots was almost consistent, and its mechanism deserves further investigation.

### 4.3. Effect of Intergeneric Grafting on Physiological Characteristics of Rootstock

Plant endogenous hormones are essential substances in vital plant activities. As extremely trace products produced by the plant's metabolism in vivo, they participate in the regulation of important physiological processes such as plant growth and development, and at the same time act as regulators of vital root activities [50]. Scions participate directly or indirectly in the establishment of root morphology by interfering with the balance of endogenous hormones in rootstocks [51]. The results showed that intergeneric grafting could decrease ZR content and increase ABA content in the rootstock, while the effects on IAA and $GA_3$ content varied with the grafted combination and the grafting period. Studies have shown that higher concentrations of cytokinin and auxin could promote the generation of adventitious roots [52], and gibberellin could affect root growth by regulating root cell division and elongation [53]. High concentrations of ABA could inhibit rooting [54]. This study found that the intergeneric grafting of Schisandraceae reduced ZR content and increased ABA content in the rootstock, while the effects on IAA and $GA_3$ content varied depending on the grafted combination and period. The results indicated that intergeneric grafting of Schisandraceae inhibited the growth of the rootstock by reducing ZR and IAA content and increasing ABA content, which was an important reason for the decrease in total root length, root volume, root tips, and root biomass of both K/S and S/K. However, the increase in $GA_3$ content did not promote the total root length, which might be that the decrease in ZR and IAA content had a greater inhibitory effect on root growth, which led to the decrease in total root length. At 70 days after grafting, IAA content in K/S suddenly increased, and the increase in $GA_3$ content reached a maximum, while a decrease in IAA content and sudden decrease in $GA_3$ content occurred in S/K. This might be the reason for the significant increase and decrease in the root average diameter of K/S and S/K, respectively, indicating that the effect of the scion on root average diameter is related to the changes in $GA_3$ and IAA content. These results also confirm that in the process of root growth and development, it is not a single hormone that regulates root morphology but the comprehensive interaction of various hormones [55].

Root activity, as a parameter that characterizes plant roots, is an important physiological index for evaluating the resistance of plant roots to stress and directly affects the nutritional status and yield of aboveground parts [50,56]. In this study, the root activity of K/S and S/K was lower than that of the control, which could be the result of the incompatibility of the intergeneric grafting compared with self-grafting, which caused adverse stress on the grafted seedlings and led to the decline in their root activity. This was similar to the results of Zhou Kaibing's research on the effect of citrus scions on the physiological characteristics of rootstocks, where the root activity of grafted rootstocks was lower than that of non-grafted rootstocks, reflecting that grafting was also a stress on the plants [17].

Malondialdehyde (MDA) is one of the main products of membrane lipid peroxidation, which increases the permeability of cell membranes, reduces their stability, and causes electrolyte leakage, leading to physiological dysfunction and aggravating membrane damage. In this study, the MDA content in S/K was significantly higher than that in K/K, indicating that intergeneric grafting caused certain stress on rootstock and reduced its resistance. This was similar to the research results that the genetic relationship between scions and rootstocks determined the MDA content. The farther the relationship, the higher the content, the stronger the stress, and the stress resistance of grafted rootstocks was lower than that of non-grafted rootstocks [7,57]. However, there was no significant difference in the MDA content of K/S compared with S/S, which indicated that the degree of stress on rootstock due to grafting might also be related to the combination of scion and rootstock. When the scion is stronger than the rootstock, the stress on the rootstock will be reduced.

Soluble sugar is one of the main substances that plants use to resist environmental stress. Maintaining high levels of soluble sugar under stress can increase a plant's ability to resist stress. Soluble proteins in plants mostly include enzymes involved in various metabolic processes. When subjected to environmental stress, these proteins undergo changes, and their levels can be measured as an important index of a plant's stress resistance [58]. The grafting experiments on grapes, oranges, and watermelons showed that the scion could significantly inhibit the content of soluble sugar and soluble protein in the rootstock [17,32,59]. Soluble sugar and soluble protein, as the material bases for cell growth [60], are also important components of nutrients required by plants. The research by Zheng Fangyi et al. showed that the root soluble sugar content and soluble protein content of non-grafted rootstocks were the highest, followed by homologous grafting, and the least being heterologous grafting, indicating that grafting reduced root nutrient storage, and the farther the genetic distance between the scion and rootstock was, the more unfavorable it was for nutrient storage [7]. In this study, the soluble sugar content of K/S and the soluble sugar and protein content of S/K were significantly lower than those of the control, consistent with the results of previous studies. This suggests that intergeneric grafting in Schisandraceae will reduce the storage of nutrients and stress resistance in rootstock. However, there was no significant difference in the soluble protein content of K/S compared to the control, indicating that the stress on the rootstock caused by *Kadsura coccinea* scion was lower than that of *Schisandra sphenanthera* scion, once again highlighting that the degree of stress on the rootstock was related to the relative strength of the rootstock and the scion.

## 5. Conclusions

This study was the first to reveal the effects of the intergeneric grafting of Schisandraceae on the rootstock in terms of morphology, anatomy, and physiology. The results showed that the intergeneric grafting of Schisandraceae caused certain stress on the rootstock, inhibited the growth of the root system and reduced its stress resistance, and affected the root morphology by regulating the content of endogenous hormones. Additionally, intergeneric grafting altered the anatomical structure of different root orders of the rootstock, thereby affecting its lateral and longitudinal transport capacity. Furthermore, the effects of intergeneric grafting on the rootstock varied with different graft combinations, which was related to the relative strength of the rootstock and scion.

**Author Contributions:** Conceptualization, S.W.; Data curation, H.-Y.L. and C.-Y.Z.; Formal analysis, H.-Y.L.; Investigation, H.-Y.L. and C.-Y.Z.; Project administration, S.W.; Resources, H.-Y.L.; Supervision, S.W.; Validation, S.W.; Writing—original draft, H.-Y.L. All authors have read and agreed to the published version of the manuscript.

**Funding:** This research was funded by the Research and Utilization of Tropical Arid Non-wood Forest Germplasm Resources in China and Pakistan (Grant No. 2022WK2021).

**Data Availability Statement:** The samples analyzed may be available upon request after a share transfer agreement. The datasets generated during the current study are available from the corresponding author upon reasonable request.

**Conflicts of Interest:** The authors declare no conflict of interest.

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
