# Peer review of "Effects of Intergeneric Grafting of Schisandraceae on Root Morphology, Anatomy and Physiology of Rootstocks"

_forests, doi:10.3390/f14061183_

Round 1

Reviewer 1 Report

Comments:

The study titled “Effects of intergeneric grafting of Schisandraceae on root morphology, anatomy and physiology of rootstocks” by Liao et al presents interesting observational data on the reciprocal scion effect on the rootstocks of Kadsura coccinea (K) and Schisandra sphenanthera (S). The study indicated that the inferior performance of the heterografts might be due to the physiological stress led by lesser compatibility of the graft partners. The manuscript is nicely written. However, it requires elaboration/clarification in some places and contains over-exaggerated texts in others. Comments have been provided below for the authors to follow and/or respond to.

Major comments:

1.     Figure 3: Root surface area (RSA) values did not appear too different from the control samples except for that of K/S at 30 DAG. The authors are suggested to modify their argument accordingly. Additionally, the authors are requested to clarify which root (primary (order 3), secondary (order 2), or tertiary (order 1)) and which region of them were used for the data retrieval.

2.     Tables 2, 3; Figures 4, 5: It is not clear which specific region was used for the anatomical data retrieval. Were all 5 mm long root samples used for the data retrieval and replicated thrice? Sectioning thickness has not been mentioned as well. Since the specific region used for sectioning may bring variation in the data retrieved, the authors are kindly suggested to clarify the aforementioned information.

3.     Figure 6: The authors apparently quantified the total IAA content of the samples, which is nice. However, quantification of its free form (free IAA) would have been more informative for the study. Authors are suggested to do so, if feasible.

4.     Subsection 2.6: The full form of TTC has not been defined (triphenyltetrazolium chloride (?))

5.     Page 13, subsection 4.1: The conclusion made at the end may be overexaggerated as the manuscript does not present any data related to the defense response/ability of the rootstock/scion. The authors are kindly suggested to modify their argument accordingly.

6.     Page 15, first paragraph- last line: The data in the study may not be sufficient to interpret the farther genetic distance as a reason behind the stress. The data from only two auto-grafts and their reciprocal hetero-grafts alone may not be sufficient in this regard. The authors are kindly suggested to modify their argument accordingly.

Minor comments:

1.     The authors are kindly suggested to refer to the roots of 1st, 2nd, and 3rd orders to tertiary, secondary, and primary roots respectively.

Author Response

Major comments:

  1. Figure 3: Root surface area (RSA) values did not appear too different from the control samples except for that of K/S at 30 DAG. The authors are suggested to modify their argument accordingly. Additionally, the authors are requested to clarify which root (primary (order 3), secondary (order 2), or tertiary (order 1)) and which region of them were used for the data retrieval.

Answer:Thank you for your question. According to your suggestion, the argument about root surface area (RSA) of K/S has been revised (subsection3.1&4.1). In addition, we clarify here that the measurement of root morphological characteristics in this study is aimed at the whole root system (mentioned in subsection2.3), including the first order roots, the second order roots and the third order roots.

  1. Tables 2, 3; Figures 4, 5: It is not clear which specific region was used for the anatomical data retrieval. Were all 5 mm long root samples used for the data retrieval and replicated thrice? Sectioning thickness has not been mentioned as well. Since the specific region used for sectioning may bring variation in the data retrieved, the authors are kindly suggested to clarify the aforementioned information.

Answer:Thank you for your questions, which have been clarified in the manuscript according to your suggestion. (subsection 2.4)

  1. Figure 6: The authors apparently quantified the total IAA content of the samples, which is nice. However, quantification of its free form (free IAA) would have been more informative for the study. Authors are suggested to do so, if feasible.

Answer:Thank you for your suggestion, because our plant materials have been used up, there are no extra experimental materials for us to further study its free IAA. We will consider adding this part of the experimental content to the subsequent related research to explore the changes of free IAA.

  1. Subsection 2.6: The full form of TTC has not been defined (triphenyltetrazolium chloride (?))

Answer:Thank you for your question. TTC is defined as triphenyltetrazolium chloride, and we have made comments in the manuscript. (subsection 2.6)

  1. Page 13, subsection 4.1: The conclusion made at the end may be overexaggerated as the manuscript does not present any data related to the defense response/ability of the rootstock/scion. The authors are kindly suggested to modify their argument accordingly.

Answer:Thank you for your suggestion, and relevant arguments have been revised in the manuscript. (subsection 4.1)

  1. Page 15, first paragraph- last line: The data in the study may not be sufficient to interpret the farther genetic distance as a reason behind the stress. The data from only two auto-grafts and their reciprocal hetero-grafts alone may not be sufficient in this regard. The authors are kindly suggested to modify their argument accordingly.

Answer:Thank you for your suggestion, and relevant arguments have been revised in the manuscript. (subsection 4.3)

Minor comments:

  1. The authors are kindly suggested to refer to the roots of 1st, 2nd, and 3rd orders to tertiary, secondary, and primary roots respectively.

Thank you for your suggestion. In this study, I quoted Pregitzer's article and used the root order grading method to classify roots. The descriptions of different root orders in related references are also the first order, the second order and the third order roots. Therefore, your suggestion will not be considered for the time being.

Reviewer 2 Report

In the tex, specify what you took for the control plants.

Explain why the temperature regime of 60°C was used when determining the biomass of roots.

In the Materials and Methods section, specify the years of research. Since it is not clear how many years the research was conducted.

Author Response

  1. In the tex, specify what you took for the control plants.

Answer:Thank you for your question. The control plants in this study are grafted seedlings S/S and K/K respectively, which has been explained in subsection 2.2.

  1. Explain why the temperature regime of 60°C was used when determining the biomass of roots.

Answer:Thank you for your question. The drying temperature of plants is usually between 40-60℃, and the roots contains a lot of water, which is generally set at 60℃, so that all the water can be evaporated without damaging the material itself. This is also based on many previous studies.

  1. In the Materials and Methods section, specify the years of research. Since it is not clear how many years the research was conducted.

Answer:Thank you for your suggestion. In the "Materials and Methods" section, the research year has been specified. (subsection 2.1)

Reviewer 3 Report

This manuscript entitled “Effects of Intergeneric Grafting of Schisandraceae on Root Morphology, Anatomy and Physiology of Rootstocks” aimed to reveal the effect of grafting between two different generic vine species on rootstock properties. Four types of grafting were conducted on K. coccinea and S. sphenantha, seedlings (K/K, S/K, S/S, and S/K), and root stock morphology, anatomical structures, and physiological characteristics were compared between within the species graftings and intergeneric graftings. The results showed that intergeneric graftings reduced rootstock sizes, enhanced cortex thickness of the second and the third order roots, varied levels of endogenous hormones in the roots and showed symptoms of stress. The study was well designed, and it is significant because it provided comprehensive information on the properties of root stocks in intergeneric grafting.

Here I make two suggestions that I believe would improve the value of this manuscript.

1. I recommend authors to append information on the growth of scion in each treatment (add in the manuscript or supplemental information), although I understand the aim of this study is to reveal the properties of rootstock in intergeneric grafting. The author discussed relative strength of the rootstock and the scion, but there is insufficient information to determine the validity of the arguments in this manuscript. I consider the information on above ground growth in this study would be useful to reveal the stress mechanism in intergeneric graftings.

2. The effect of interaction among endogenous hormones on root growth was discussed in subsection 4.3. I wonder this point might be clarified by comparing the number and size of cells in root cross-section between K/S (S/K) and S/S (K/K) or between S/K and K/S.

Minor remarks

Abstract

L14 “Anatomically, root cross section…”  and L15 “And cortex thickness…” seems to contradict each other in description about cortex thickness of S/K.

Materials and Methods

2.5

L13: “GA3” “GA3

2.9

In this study, all statistical comparisons were conducted on between two mean values, hence there is no need to conduct multiple comparison (Duncan’s test).

Results

3.4

L5:  “CK” “S/S” ?

L6: “CK” “K/K” ?

Figures and Tables

AS pointed out above, all statistic comparisons were conducted on between two values. Hence, I consider that asterisking pairs with significant differences is more suitable representation than letter attaching (simpler and easier to confirm).

Note in Table 2 and 3

“Means provided with error bars” “Means provided with standard errors”

The lowest Left label in Figure 5

“S/K”   “K/K” ?

Author Response

Major remarks:

  1. I recommend authors to append information on the growth of scion in each treatment (add in the manuscript or supplemental information), although I understand the aim of this study is to reveal the properties of rootstock in intergeneric grafting. The author discussed relative strength of the rootstock and the scion, but there is insufficient information to determine the validity of the arguments in this manuscript. I consider the information on above ground growth in this study would be useful to reveal the stress mechanism in intergeneric graftings.

Answer:Thank you for your suggestion. Information about the growth of scions has been explained in the manuscript (subsection 2.2), and the differences between Kadsura coccinea and Schisandra sphenanthera can also be observed from Figures 1A&1B.

  1. The effect of interaction among endogenous hormones on root growth was discussed in subsection 4.3. I wonder this point might be clarified by comparing the number and size of cells in root cross-section between K/S (S/K) and S/S (K/K) or between S/K and K/S.

Answer:Thank you for your suggestion. Because the paraffin section made in this experiment can't accurately calculate the number and size of each cell in the root cross section under microscopic observation, the relevant values are not counted in the results, so the relationship between the number and size of cells and endogenous hormones can't be analyzed in the discussion. We will further refine the paraffin section technique in the follow-up study to obtain more comprehensive anatomical parameters.

Minor remarks:

  1. L14 “Anatomically, root cross section…” and L15 “And cortex thickness…” seems to contradict each other in description about cortex thickness of S/K.

Answer:Thank you for your reminder. I'm sorry that we really made a mistake and didn't check carefully when submitting the manuscript. We have revised it in the manuscript, and L14 “cortex thickness" has been deleted..

  1. Materials and Methods-2.5-L13: “GA3” → “GA3”

Answer:Thank you for your reminder. It has been revised in the manuscript according to your instructions.

  1. Materials and Methods-2.9-In this study, all statistical comparisons were conducted on between two mean values, hence there is no need to conduct multiple comparison (Duncan’s test).

Answer:Thank you for your suggestion.

  1. Results-3.4-L5: “CK” → “S/S” ? & L6: “CK” → “K/K” ?

Answer:Thank you for your suggestion. It has been revised in the manuscript according to your instructions.

  1. AS pointed out above, all statistic comparisons were conducted on between two values. Hence, I consider that asterisking pairs with significant differences is more suitable representation than letter attaching (simpler and easier to confirm).

Answer:Thank you for your suggestion. It has been revised in the manuscript according to your instructions.

  1. Figures and Tables-Note in Table 2 and 3-“Means provided with error bars” → “Means provided with standard errors”

Answer:Thank you for your reminder. It has been revised in the manuscript according to your instructions.

  1. Figures and Tables-The lowest Left label in Figure 5-“S/K” → “K/K” ?

Answer:Thank you for your reminder. It has been revised in the manuscript according to your instructions.

.
